# The Roles of Nitric Oxide Synthase/Nitric Oxide Pathway in the Pathology of Vascular Dementia and Related Therapeutic Approaches

**DOI:** 10.3390/ijms22094540

**Published:** 2021-04-26

**Authors:** Han-Yan Zhu, Fen-Fang Hong, Shu-Long Yang

**Affiliations:** 1Department of Physiology, College of Medicine, Nanchang University, 461 Bayi Avenue, Nanchang 330006, China; zhu626314308@foxmail.com; 2Queen Marry College, College of Medicine, Nanchang University, 461 Bayi Avenue, Nanchang 330006, China; 3Teaching Center, Department of Experimental, Nanchang University, 461 Bayi Avenue, Nanchang 330006, China

**Keywords:** vascular dementia, nitric oxide synthase, nitric oxide, pathology, therapeutic approaches

## Abstract

Vascular dementia (VaD) is the second most common form of dementia worldwide. It is caused by cerebrovascular disease, and patients often show severe impairments of advanced cognitive abilities. Nitric oxide synthase (NOS) and nitric oxide (NO) play vital roles in the pathogenesis of VaD. The functions of NO are determined by its concentration and bioavailability, which are regulated by NOS activity. The activities of different NOS subtypes in the brain are partitioned. Pathologically, endothelial NOS is inactivated, which causes insufficient NO production and aggravates oxidative stress before inducing cerebrovascular endothelial dysfunction, while neuronal NOS is overactive and can produce excessive NO to cause neurotoxicity. Meanwhile, inflammation stimulates the massive expression of inducible NOS, which also produces excessive NO and then induces neuroinflammation. The vicious circle of these kinds of damage having impacts on each other finally leads to VaD. This review summarizes the roles of the NOS/NO pathway in the pathology of VaD and also proposes some potential therapeutic methods that target this pathway in the hope of inspiring novel ideas for VaD therapeutic approaches.

## 1. Introduction

Population aging is a global phenomenon. In 2017, the aging population in Asia had reached approximately 365 million, and it is estimated to reach approximately 520 million by 2030 [1,2]. Dementia is the most common disease that develops as the body ages. The World Health Organization (WHO) estimates that it will increase to nearly 9.9 million new cases each year [3,4]. Dementia leads to a progressive decline in cognitive abilities. Since 2012, the WHO listed dementia as one of the main threats to the world’s population [5]. Vascular dementia (VaD) is the second most common dementia after Alzheimer’s disease (AD), accounting for about 20–30% of all dementia cases [6]. Since the development of VaD is closely related to cardiovascular and cerebrovascular diseases, it has a higher mortality rate than AD, with an average survival time of three-to-five years [7,8]. Compared with AD and other neurogenerative diseases, cognitive impairments in VaD appear more variable. Patients often suffer from attention-deficit disorder, forgetfulness, and severe impairments of high-level cognitive abilities, such as processing and judgment ability. Additionally, depression and apathy are particularly prominent in VaD patients [9,10,11]. These symptoms can not only greatly reduce patients’ living quality but also cause great pain to their family and impose a huge burden on society and the economy [12].

The definition of VaD has not been integrated yet. However, the concept of vascular cognitive impairment (VCI), i.e., “a syndrome with evidence of clinical stroke or subclinical vascular brain injury and cognitive impairment affecting at least 1 cognitive domain” has been introduced. It encompasses all the cognitive disorders associated with cerebrovascular disease, and it covers various symptoms ranging from mild cognitive impairment to dementia. [13,14]. VaD is the most serious form of VCI. Currently, the Diagnostic and Statistical Manual of Mental Disorders—5th Edition (DSM—V) [15], Vascular Behavioral and Cognitive Disorders (Vas–Cog) [16], and neuroimaging [17] are mainly used for the clinical diagnosis of VaD.

The presence of abnormal cerebrovascular pathology doubles the incidence of dementia, which develops from neurodegenerative diseases [18]. Cerebral small vessel disease (CSVD) is considered to be the most important cause of VaD. CSVD can cause arteriolosclerosis, lacunar infarcts, cortical and subcortical microinfarction, and diffuse white matter (WM) lesions [13,19]. The majority of VaD cases are manifested as subcortical ischemia and hypoxia injury. [20,21]. The accumulation of damage to the subcortical vascular system can reduce cerebral blood flow (CBF) and then lead to changes in vascular integrity and chronic hypoperfusion (especially in the basal ganglia, WH, and hippocampus). Subsequently, cerebrovascular endothelial cell (CEC) dysfunction develops and further reduces CBF [22,23].

Stroke, atherosclerosis, diabetes, obesity, and hypertension are common risk factors for VaD [24,25,26]. These diseases keep the body in a state of chronic inflammation, thereby deteriorating the function of subcortical blood vessels, causing neuroinflammation, which then causes WM damage (e.g., demyelination and axon loss), and eventually, dementia [13,22,27].

Nitric oxide synthase (NOS) and nitric oxide (NO) are important signal molecules that regulate the nervous and vascular system, and have a variety of physiological and pathological effects [28]. Physiologically, NO acts as an endogenous vasodilator and important intercellular messenger in the cerebral and peripheral blood flow [20,23]. In the central nervous system, NO participates in the development of cognitive functions [29]. However, NO is an active free radical gas with strong oxidizing abilities. While, pathologically, NO can react with superoxide anions (O_2_^−^) to form strong oxidant peroxynitrite (ONOO^−^) [30,31]. ONOO^−^ can trigger oxidative stress and then cause cell damage, such as protein degeneration and inactivation, lipid peroxidation, and DNA damage that further aggravate pathological damage of the body [32]. Oxidative stress is an environment caused by an imbalance of oxidants and antioxidants in the body, which is manifested by the massive accumulation of harmful free radicals and superoxide [33]. Oxidative stress has been proven to be widespread in cognitive or vascular diseases such as AD [34], hypertension [35], and atherosclerosis [36]. All risk factors of VaD can induce oxidative stress, thereby causing NOS/NO pathway alterations, and the damaged NOS/NO pathway can, in turn, aggravate oxidative stress and ultimately lead to a series of pathological changes in VaD [32,37].

NO production can be catalyzed by NOS. In mammals, NOS has three main isoenzymes: endothelial NOS (eNOS), neuronal NOS (nNOS), and inducible NOS (iNOS). They have different characteristics and produce NO with different rates [28]. eNOS is mainly expressed by endothelial cells, and eNOS-derived NO is closely related to the function of CECs and the regulation of CBF [38]. nNOS is expressed in central and peripheral neurons, and nNOS-derived NO contributes to the formation of learning and memory by affecting synaptic plasticity [39]. Though iNOS is almost not expressed in healthy states, it can be induced or stimulated by various cytokines such as tumor necrosis factor-α (TNF-α), tumor necrosis factor-β (TNF-β), and interleukin-6 (IL-6). Under inflammatory conditions after immunological or microbial stimulation, iNOS can be expressed and remain active to produce massive NO in order to mediate inflammation [40,41]. These three pathways have been proven to play an indispensable role in the cerebrovascular dysfunction and cognitive dysfunction that occur in AD [42], stroke [43], and atherosclerosis [44].

Though the number of patients suffering from VaD keeps increasing, the treatment of VaD is still pretty limited. Currently, there are no approved VaD-specific drugs, except for some drugs for the treatment of AD, such as donepezil and memantine used in clinics. However, these drugs can only marginally and temporarily control VaD symptoms, and they may induce side effects, like delirium in VaD patients [14,45,46,47].

The neurovascular protective effects of the NOS/NO pathway have been founded in healthy individuals, so clarifying the way they alternate in VaD can greatly help us understand its pathological mechanisms. This could provide plenty of new ideas for the development of new drugs or treatments, as well as identify more potential biomarkers to diagnose or observe the disease progression. This review focuses on the roles of the NOS/NO pathway in the pathogenesis of VaD, and it summarizes some of the therapies targeting this pathway for VaD treatment.

## 2. The Pathogenesis of VaD

The brain injuries of most VaD patients are manifested as diffuse WM damage and cerebral infarction in brain imaging [14,17]. Studies have found that these brain injury areas are under chronic hypoxia, with unstable hemodynamics, attenuated vascular reactivity, and reduced CBF [21,48]. These injuries are specifically manifested as CEC dysfunction, nervous impairment, and inflammation, which can mutually influence each other and further promote VaD development (Figure 1).

### 2.1. VaD and Neurovascular Dysfunction (CEC Dysfunction and Nervous Impairment)

The main vascular cells in the brain are CECs, vascular smooth muscle cells (VSMCs), and pericytes. They are all closely connected with other brain cells, such as neurons and glial cells. They jointly build brain structure and regulate brain functions. These cells form a dynamic functional domain called the neurovascular unit (NVU) [49]. As the fundamental pathology of VaD, CEC dysfunction and nervous impairment (collectively known as neurovascular dysfunction) comprise essentially structural and functional damage of the NVU. Plenty of studies have already proven that the NVU is severely damaged in VaD [50,51,52].

CECs are the most crucial component of the NVU—they can release various vasoactive agents including vasoconstriction factors (e.g., endothelin-1 and thromboxane A2) and vasorelaxation factors (e.g., NO and prostaglandin E2) to regulate CBF supply by controlling vascular tone [53]. CECs can also form tight junctions to maintain the integrity of the blood–brain barrier (BBB) [54,55].

CEC dysfunction is especially obvious in the brains of the elderly [56,57,58]. It leads to the decline of CBF and oxygen supply, and then induces disorders in the other components of the NVU. Therefore, CBF is unable to respond to nerve signals, resulting in an insufficient energy supply to the nervous system. Additionally, CEC dysfunction can trigger BBB damage, which induces inflammation and neurotoxicity. Later, neuronal impairments or death occurs, eventually leading to cognitive dysfunctions [54,56,59].

### 2.2. VaD and Inflammation

All VaD risk factors, such as stroke and atherosclerosis, can cause chronic inflammation in animal models and humans [19,52,60]. Additionally, cell senescence has been found to express a unique pro-inflammatory phenotype. That is to say, senescent cells can secrete inflammatory factors such as TNF-α, TNF-β, and cyclooxygenase-2 (COX-2) [61]. Researchers found that the TNF-α concentration in the CECs of old mice was much higher than that of young mice [62]. A large number of studies have shown that, compared with healthy controls, elevated levels of inflammatory cytokines are detected in the damaged WM of VaD patients [60,63,64]. All of these results suggest the significant effect of inflammation in VaD pathology. To be more specific, senescence and VaD risk factors lead to CEC dysfunction, which increase the BBB’s permeability. As a result, inflammatory cytokines and cytotoxic molecules can more easily invade the brain. After that, inflammation can trigger the exudation of plasma proteins, cause tissue edema to compress blood vessels, and further aggravate CEC dysfunction. Additionally, the massive release of inflammatory factors can cause neuroinflammation, thereby continuously leading to neuronal damage or even death. The nervous impairments combined with the persistent decline of CBF increase the area of cerebral infarction, deteriorate cognitive abilities, and eventually lead to VaD [65,66,67] Correspondingly, many studies have found that some treatments (such as simvastatin [68], omega-3 PUFAs [69], and acupuncture [70]) can remarkably attenuate WM damage and even cognitive impairments in VaD mice by inhibiting inflammation.

The basic pathological mechanisms of VaD are neurovascular dysfunction (CEC dysfunction and nervous impairment) and inflammation. Risk factors and aging cause CBF to decrease, resulting in oxidative stress, hypoxia, and hypoperfusion in brain tissues, thus inducing CEC dysfunction and inflammation. Moreover, CEC dysfunction and inflammation can exacerbate each other. Additionally, inflammation can aggravate oxidative stress and then produce cytotoxicity, which can lead to neuroinflammation. CEC dysfunction and neuroinflammation can seriously damage the NVU and weaken the response of CBF to neural signals, causing the continuous decline of CBF. These changes interact with each other and continue to worsen, leading to severe damage of the WM and hippocampus, inducing cognitive impairment, and finally developing VaD.

## 3. NOS/NO Pathway Overview

NO is a messenger molecule with numerous molecular targets, and it can mediate a variety of biological functions. Since NO is uncharged and highly soluble in a hydrophobic environment, it can diffuse freely in cell membranes [71,72].

NO synthesis mainly has two steps. Firstly, NOS hydroxylates l-arginine (L-Arg) to *N*-omega-hydroxy-l-arginine (NOHA). Secondly, NOS oxidizes NOHA to l-citrulline and NO. This enzymatic reaction requires substrates such as L-Arg and O_2_ and cofactors such as reduced nicotinamide-adenine-phosphate dinucleotide (NADPH), flavin adenine dinucleotide (FAD), flavin mononucleotide (FMN), and tetrahydrobiopterin (BH_4_) [73].

NOS is only functional as dimeric structure (Figure 2). The monomeric NOS is inactive, which can bind neither cofactor BH_4_ or substrate L-Arg [73,74]. Normally, the oxygenase domain at the N terminal is combined with Heme to gain the activity of binding cofactors. Additionally, FAD and FMN attached to the reductase domain at the C terminal can transfer an electron provided by NADPH oxidase to Heme. The two domains are connected by calmodulin (CaM), which can enhance the electron transfer from NADPH oxidase [73,74,75,76]. Thus, when there is sufficient substrate L-Arg and cofactor BH_4_ at the Heme site, the previously obtained electrons can catalyze the formation of NO from L-Arg and O_2_ [72,73].

There are three subtypes of NOS in mammals: eNOS, nNOS, and iNOS. In healthy individuals, NO is only produced by eNOS in endothelial cells and nNOS in the nervous system. eNOS and nNOS are collectively called constitutive NOS (cNOS). The combination of cNOS and CaM is dependent on the concentration of Ca^2+^. Thus, the rapid influx of Ca^2+^ induced by tissue damage can activate cNOS with fast responses to play a protective role. On the contrary, the combination of iNOS and CaM does not require Ca^2+^ regulation, and iNOS is only expressed under the induction of inflammation [31,77,78].

The monomeric form of NOS is inactive, and only after the formation of a functional dimeric structure does NOS bind the cofactor BH_4_ and the substrate L-Arg to catalyze NO formation. In the dimeric form, the N-terminal catalytic domain of the enzyme binds Heme and CaM, which gives the enzyme the activity to bind BH_4_. At the same time, there are FAD and FMN in the C-terminal reductase domain, which can speed up the transfer of an electron provided by NADPH oxidase to the Heme domain. After that, with the help of cofactor BH_4_, the complete NOS dimeric structure can catalyze the formation of L-Arg into NO.

## 4. NOS/NO Pathway Plays a Key Role in Various Pathogenesis of VaD

### 4.1. NOS/NO Pathway Exhibits Dual Functions of Protection and Damage in the Brain

The brain is the organ with the highest NOS activity in the human body. Plenty of studies have shown that NO has both neuroprotective and neurotoxic tendencies, depending on its concentration, bioavailability, and the redox state of the tissue [71,72]. Thus far, NO plays at least five significant roles in the brain: (1) it can be a vasoactive factor for endothelial relaxation and regulating CBF; (2) it can act as a neurotransmitter, increasing the level of the second messenger cyclic guanosine monophosphate (cGMP) to regulate the tension of vascular smooth muscle; (3) it can act as an appropriate concentration of NO can reduce excessive calcium ion (Ca^2+^) influx which caused by injury and can protect nerves; (4) an excessive amount of NO plays cytotoxic effect and leads to nervous impairments or even death and; (5) it can participate in the development and maintenance of learning and memory functions [31,32,79,80,81].

Some studies have detected changes in the expression and activity of NO metabolites, donors, and NOS/NO itself in the brains of AD and VaD patients [82,83,84]. A review on animal stroke models showed that a supplemented NO donor improves overall CBF and reduces infarct size [85]. Moreover, researchers have found that mice with eNOS gene knockout have larger infarcts than wild-type ones after middle cerebral artery occlusion [84,86], while mice with nNOS or iNOS gene knockout have smaller infarcts [87,88]. These results suggest that these three NOS subtypes play completely different roles in their respective regions and collectively affect the functions of brain. Currently, it has been proven that alterations of the NOS/NO pathway in VaD pathology manifest as a decrease of eNOS activity and the excessive activation of nNOS and iNOS [79,82].

### 4.2. Oxidative Stress Is a Vital Mechanism of NOS/NO Pathway to Cause VaD

Free radicals, such as reactive oxygen species (ROS) and reactive nitrogen species (RNS), are normal cell metabolic products that participate in a variety of cellular events such as signal transduction in the BBB and enzyme catalysis [89]. These agents usually have at least one unpaired electron, so they are quite active and easily react with various substances in the body to produce O_2_^−^ and peroxides. Meanwhile, distributing multiple antioxidants, such as superoxide dismutase (SOD), glutathione peroxidase (GPx), and glutathione reductase (GR), onto healthy individuals can neutralize the strong oxidizing property of free radicals to maintain homeostasis [33,90].

However, under the effects of aging [91] and vascular diseases (such as stroke [92] and atherosclerosis [93]), free radicals excessively accumulate and trigger oxidative stress. Many studies have found oxidative stress in VaD pathology. For example, Zhang et al. [94] and Gocmez et al. [95] found that SOD and GPX levels were significantly reduced in the serum of VaD mice; Casado et al. [96] detected an elevated level of malondialdehyde (MDA), a marker of lipid peroxidation damage, in the blood of VaD patients; and Gackowski et al. [97] detected increases of 8-oxo-2-deoxyguanosine and 8-oxoguanine, indicators of DNA oxidative damage, in the urine and cerebrospinal fluid (CSF) of VaD patients. Moreover, these phenomena can become more obvious along with aging.

ROS generates O_2_^−^ in an oxidative stress environment. The reaction speed of O_2_^−^ with NO is three-to-four times that of SOD. Thus, when the concentration of O_2_^−^ increases or the level of NO is excessive, they can react to produce a large amount of the cytotoxic substance ONOO^−^ [98]. ONOO^−^ can easily penetrate into cells membranes and perform nitrosylation via tyrosine and cysteine residues, resulting in oxidative damage to biological macromolecules, especially lipids, proteins, and DNA [91,94,99]. Oxidative stress consumes the protective NO and reduces NO bioavailability by generating ONOO^−^, which, in turn, aggravates oxidative stress and forms a vicious circle [99]. Since the oxygen utilization rate and lipid content of the human brain are higher than in other tissues, the brain is more susceptible to oxidative damage [100].

Oxidative stress is considered to be a vital mechanism of the NOS/NO pathway to cause VaD, especially triggering endothelial cell dysfunction. Bhayadia et al. [101] found that compared with young mice, the endothelial-dependent vasodilation (EDD) of elderly mice arteries significantly decreased. After applying nitroglycerin (a type of NO donor), the vasodilation of old mice was found to significantly recover. However, after applying L-NAME (a type of NOS inhibitor), EDD almost disappeared in both young mice and elderly mice after treatment. When anti-oxidant therapy was applied to elderly mice, if L-NAME was applied at the same time, the treatment could also be ineffective. These results suggested that there is a strong correlation between oxidative stress-induced cell senescence and endothelial dysfunction, which is NO-dependent. Moreover, the decrease of NO bioavailability, the alteration of NOS expression and activity have been found to extensively occur in the course of aging-related diseases, especially VaD [31,79,82]. All the above-mentioned experimental results suggest that the NO/NOS pathway is closely related to vascular homeostasis and participates in a variety of age-induced pathological activities. The changes of the NOS/NO pathway regulated by the three subtypes of NOS in VaD pathology are described in the following section.

## 5. NOS/NO Pathway and Neurovascular Dysfunction

In healthy individuals, eNOS and nNOS can produce certain levels of NO to maintain the integrity of the NVU and regulate brain homeostasis. eNOS-derived NO is an important mediator for maintaining vascular tone and CBF self-regulation [31,57], while nNOS-derived NO regulates neural signal transmission and maintains cognitive abilities [102]. However, under pathological conditions, the production of NO is changed and the bioavailability of NO decreases. These alterations cause serious adverse changes in neurovascular functions, including vasoconstriction, high arterial blood pressure, VSMC proliferation, platelet aggregation and adhesion, leukocyte adhesion, and nerve cells apoptosis [103,104,105,106]. NVU dysfunction is considered to be the pathological basis for the development of VaD. This process is described below from two aspects: CEC dysfunction and nervous impairment.

### 5.1. CEC Dysfunction (eNOS-Dominated)

A recent study observed that, in patients who were exposed to vascular risk factors, EDD dysfunction could be detected before the morphological changes in vascular walls, which suggests that the vascular endothelium is the first to be damaged [107]. Moreover, when the brain is under oxidative stress, CECs have been confirmed to be the most susceptible sites [100].

CEC dysfunction may be the key initiative event of VaD and to be sufficient to cause BBB damage and NVU destruction [61,106,107]. Fujita et al. [108] found that bone marrow mononuclear cell (BMMNC) therapy can effectively alleviate ischemic WM injury in model mice by significantly increasing CBF. However, after applying eNOS inhibitors, researchers found that the recovery effect of BMMNC on CBF was greatly inhibited. While this did not occur when nNOS inhibitors are applied. These results suggested that the pathological changes of cerebral ischemia may be eNOS-dependent.

Interestingly, whether the level of eNOS mRNA decreases in senescent cells is unclear [109,110,111], but eNOS activity is known to decrease with age [79]. To explain that, oxidative stress may be more likely to cause a lack of eNOS substrates or cofactors, or it may change the phosphorylation status of eNOS [58,109].

#### 5.1.1. eNOS Uncoupling

Due to a lack of substrates or cofactors, eNOS reacts with oxygen to produce O_2_^−^ instead of catalyzing L-Arg to produce NO. This situation is called eNOS uncoupling (Figure 3). The cofactor BH_4_ is necessary for eNOS to combine with a substrate, and then NO can be synthesized. BH_4_ can also stabilize the dimeric structure of eNOS [110,111]. However, BH_4_ is oxidized to dihydrobiopterin (BH_2_) by ROS, which leads to a lack of cofactors and induces eNOS uncoupling to produce O_2_^−^. At the same time, vascular risk factors enhance the activity of NADPH oxidase in the vascular wall, which also produces O_2_^−^. Subsequently, the remaining NO in the body react with O_2_^−^ to form ONOO^−^, which can induce cytotoxicity. ONOO^−^ can also oxidize BH_4_ to trihydrobiopterin (BH3), which further represses BH_4_ to aggravate eNOS uncoupling. Furthermore, ONOO^−^ can oxidize soluble guanylate cyclase (sGC) and reduce its reactivity to NO. It reduces the production of cGMP, preventing cGMP-dependent kinase from being activated and failing to carry out vasodilation [59,71,94,111,112].

Additionally, recent studies have observed that supplementation with BH_4_ can only partially restore the functions of eNOS, suggesting that there is another potential mechanism of eNOS uncoupling. Investigations have found that in hypertensive animals, the weakening of EDD is accompanied by an increase in eNOS *S*-glutathionylation. There are two highly conserved cysteine residues in the reductase domain of eNOS. Under oxidative stress, the two cysteine residues can both bind to glutathione, thereby triggering eNOS *S*-glutathionylation [113,114], which can also induce eNOS uncoupling to produce O_2_^−^.

Interestingly, some researchers have found that the above-mentioned eNOS uncoupling is not structural uncoupling, and it does not destroy the dimer enzyme structure to produce monomers. The great increase in eNOS monomers found in old arteries in the previous studies was actually an increase in endogenous IgG. After removing the IgG signal, Chang et al. [115] found that the eNOS monomer level does not increase with age. Studies have also found that compared with eNOS monomers, dimers have a stronger ability to generate superoxide [116], and once dimers are formed, it is difficult for them to return to the monomer form [117]. Additionally, different treatments of cell lysates can significantly affect the relative contents of various eNOS forms [115], which may have been the cause of errors in previous experimental results. The above-mentioned results all suggested that the uncoupling of eNOS in VaD pathology is probably only functional uncoupling. Therefore, eNOS activity is a worthy further study direction. For those results that showed changes in eNOS monomer content, further careful analysis may be required.

Under the action of oxidative stress, (1) ROS can oxidize cofactor BH_4_ into BH_2_, which cannot bind to eNOS and lead to eNOS uncoupling, and (2) directly trigger eNOS *S*-glutathionylation and then uncoupling. Uncoupling causes eNOS to synthesize O_2_^−^ instead of NO. Moreover, vascular risk factors can upregulate NADPH oxidase, which can also produce O_2_^−^. O_2_^−^ can react with NO to produce ONOO^−^, which further reduces the NO content in the body. ONOO^–^ can transform BH_4_ into BH_3_, which also cannot bind to eNOS and aggravate eNOS uncoupling. Furthermore, ONOO^−^ can induce cytotoxicity and damage the vasodilator function of NO, finally reducing the bioavailability of NO.

#### 5.1.2. Changes in Phosphorylation of eNOS

Normally, eNOS is not activated, and its calcium-binding domain binds with caveolae 1 (Cav-1). In the period when the body can also repair itself, such as the early stage of cerebral ischemia, CECs can release vasoactive factors, such as vascular endothelial growth factor (VEGF), and generate shear stress to induce Ca^2+^ rapid influx and activate CaM. This can lead to the dissociation of eNOS from Cav-1, and then eNOS can combine with CaM to form an eNOS/CaM complex. Subsequently, the complex is transferred into the cytoplasm and binds with cofactors to fully activate eNOS [58,118].

eNOS phosphorylation can effectively regulate the above-mentioned mechanism. Changes in phosphorylation status at the Thr495 and Ser1177 loci of eNOS have been proven to have the greatest impact on eNOS activity. That is to say, phosphorylation at the Thr495 locus can inhibit the binding of eNOS and CaM, thereby reducing eNOS activity, while phosphorylation at the Ser1177 locus can stabilize the binding of eNOS and CaM, thereby enhancing eNOS activity [119,120,121].

The phosphatidylinositol-3-kinase (PI3K)/serine threonine protein kinase B (Akt) pathway plays an important role in the regulation of eNOS phosphorylation (Figure 4). In the early stage of cerebral ischemia, PI3K can activate Akt in the cytoplasm and then induce the phosphorylation of the eNOS/CaM complex at the Ser1177 locus and the dephosphorylation at the Thr495 locus, which can promote the activation of eNOS to exert a vascular protective effect [108,118,122]. However, in severe cerebral ischemia, the brain loses its ability to self-regulate and is under severe oxidative stress. The harmful substances produced by oxidative stress such as ROS and oxidized low-density lipoproteins (ox-LDL) can inhibit the PI3K/Akt pathway. As a result, the binding of eNOS to heat shock protein 90 (HSP90) is disrupted, and the phosphorylation of eNOS Ser1177 is inhibited, leading to eNOS being unable to dissociate from Cav-1 [36,123,124]. This leads to the inactivation of eNOS and even the uncoupling of eNOS, which can aggravate CEC dysfunction.

The dephosphorylation of eNOS Ser1177 and eNOS uncoupling are often observed in VaD patients’ brains. Researchers have found that protein tyrosine phosphatase (an inhibitor of the PI3K/Akt pathway) inhibitors can significantly improve the vascular functions and cognitive abilities of VaD mice. They have also detected an increase in NO metabolites and eNOS activity in these mice, suggesting that the phosphorylation status of eNOS is also an important cause of CECs dysfunction [119,121,125,126].

The above-mentioned mechanism has been found to be the basic pathogenic mechanisms of estrogen-deficient VaD. The vascular protective effect of estrogen is reflected in the rapid activation of eNOS through the PI3K/Akt pathway. These active eNOS can produce adequate NO in CECs to mediate the vasodilation response. Hence, post-menopausal women with estrogen deficiency are more susceptible to VaD because eNOS cannot be normally activated through the PI3K/Akt pathway [28,127,128]. Additionally, insulin resistance (IR) in diabetic patients is closely related to dementia [129,130]. The PI3K/Akt pathway is involved in various insulin responses, and IR can reduce the activity of this pathway, which can affect eNOS phosphorylation and cause CEC dysfunctions. Resveratrol, an antioxidant drug, has been found to restore CEC functions in patients with type 2 diabetes by preserving the phosphorylation of eNOS at Ser1177 [131].

The phosphorylation status of eNOS can be regarded as an effective therapeutic target of VaD. However, how to balance the phosphorylation of Thr495 and Ser1177 is an urgent problem that needs to be solved. Lee et al. [132] found that bupivacaine (BPV) can only induce Ser1177 phosphorylation at high concentration (150 μM) and manifest as vasodilatory effects. At low concentration (30 μM), BPV was found to be able to induce the phosphorylation of Thr495 and Ser1177 simultaneously, as well as manifest as vasoconstrictive effects. However, the reason why these occur is not clear. In addition, studies have found that there is an autoinhibitory element in the FMN-binding subdomain of eNOS that can regulate the phosphorylation of Thr495 and Ser1177, thereby maintaining eNOS activity [133]. Therefore, the specific molecular mechanisms of the therapeutic effects of related drugs still need to be further studied.

#### 5.1.3. eNOS and Angiogenesis

The eNOS/NO pathway has also been found to be the major mediator of angiogenesis after ischemia. Some studies have found that eNOS uncoupling can reduce the number and function of endothelial progenitor cells (EPCs), thereby preventing the synthesis of new CECs [134]. The eNOS/NO pathway can regulate the production of brain-derived neurotrophic factor (BDNF). BDNF is a type of neurotrophic factor and plays an essential role in nerve production and survival (this is described in detail in the next section) [135]. Recently, Descamps et al. [136] found that supplementation with exogenous BDNF could induce the formation of vascular endothelial cells. They further found that this is mediated by transcriptional repressor enhancer of zeste homolog 2 (EZH2) and microRNA-214 (miR-214). EZH2 can occupy the eNOS promoter, thereby inhibiting eNOS transcription, while BDNF can increase miR-214 expression, which can inhibit EZH2 expression and then increase the production of eNOS. Therefore, under the effects of aging and VaD risk factors, the signal of the PI3K/Akt pathway is downregulated, leading to eNOS uncoupling, which can decrease eNOS activity [100]. Thus, BDNF cannot be synthesized, resulting in a decrease in miR-214 expression and an increase in EZH2 expression. EZH2 can decrease eNOS expression. Thus, the eNOS/NO pathway is further damaged, resulting in EPC dysfunction or even the failure of EPC synthesis—eventually seriously affecting angiogenesis.

A recent study found that limb remote ischemic conditioning (LRIC) can generate new blood vessels in the hippocampus of VaD model mice, thereby improving chronic cerebral hypoperfusion. However, after applying eNOS inhibitors to these treated mice, these improvements were found to be significantly repressed [137]. This indicated that eNOS/NO pathway damage may lead to angiogenesis dysfunction, which makes the brain fail to repair itself. These findings suggested that angiogenesis can be a promising strategy for VaD treatment.

### 5.2. Nervous Impairment

#### 5.2.1. nNOS and Neurovascular Coupling

nNOS has been confirmed by many studies to have great significance in multitudinous neural signal events. nNOS-derived NO plays an essential role in learning, memory, neurogenesis, and synaptic plasticity [104,138]. Neurovascular coupling (NVC) is a fundamental mechanism for NVU cells to maintain brain homeostasis. That is to say, after NVU cells respond to nerve signals, they regulate local CBF distribution to meet the energy needs of different brain regions. It is a kind of spontaneous functional hyperemia that plays a vital role in maintaining brain neurological functions such as cognition [50]. When the CBF is insufficient or the signal response is weakened, NVC dysfunction occurs, as can be found in the aging process of humans and mice [59,139].

nNOS in the nitrergic nerve produces NO as neurotransmitters that can stimulate effector cells to produce NO-sensitive sGC. Subsequently, sGC can activate cGMP to reduce the tension of vascular smooth muscle and eventually help the NVC work properly [102]. However, under pathological conditions, such as CEC dysfunction and inflammation, a large amount of Ca^2+^ influx is stimulated, which can induce the nitrosylation of *N*-methyl-d-aspartic acid (NMDA) and enhance its excitability. As a result, nNOS is over-activated, and then it produces excessive NO, which can interact with O_2_^−^ to form ONOO^−^. Large amounts of ONOO^−^ can aggravate oxidative stress and induce strong neurotoxicity, which can lead to nerve damage and even mass death [140,141,142]. Excessive NO production has been found to exist throughout all stages of VaD, which can reduce CBF responses to normal signals and lead to severe impairment of cognition [50,83]. A recent study found that the cortex of mice that with nNOS gene knockout showed resistance to neurotoxicity [143].

#### 5.2.2. eNOS and Neurotrophic Factors

BDNF can support neuron survival by stimulating the proliferation and differentiation of neuron precursors and facilitating the vascular migration of neuroblasts [135,144,145]. Moreover, BDNF can induce the vascular protective effects of astrocytes, such as participating in the formation and maintenance of BBB [146,147,148]. As mentioned above, the production of BDNF is dependent on the eNOS/NO pathway. Studies have shown that CECs can produce BDNF 50 times faster than cortical neurons [149], and there is a positive feedback loop between BDNF and eNOS-derived NO production [150]. Eguchi et al. [151] found that low-intensity pulsed ultrasound (LIPUS) could effectively recover the cognitive abilities in VaD mice. In these treated mice, they detected many neurologically beneficial signs, such as an increase of neurotrophic factors. However, in treated mice with eNOS gene knockout, these signs could not be detected and their VaD symptoms did not improve. Many therapies that are used to treat endothelial dysfunction, such as simvastatin [152] and repetitive transcranial magnetic stimulation (rTMS) [153], can increase the level of BDNF in AD or VaD patients’ brains and improve their cognitive abilities. These results indicated that the reduction of BDNF synthesis, which is caused by the damage of the eNOS/NO pathway, also play an important role in the pathology of VaD.

Synaptic plasticity is considered to be the neurobiological basis of cognitive abilities and memory storage. Long-term enhancement (LTP) is one of the typical enhancement types that is closely related to long-term memory [58,107]. When nerve synapses are reduced, brain energy is abnormally consumed, thereby significantly declining CBF and ultimately aggravating neurovascular dysfunction [68]. There have been abundant studies confirming that LTP is impaired in VaD patients and animals. Additionally, in eNOS gene knockout animals, it is very difficult to form LTP [154,155,156]. The pathological mechanisms of nervous impairment caused by the eNOS/NO pathway can be briefly described as follows: eNOS/NO damage that is caused by oxidative stress can reduce the synthesis of BDNF. Insufficient BDNF prevents neuroblasts from migrating to damaged areas, thereby hindering nerve regeneration to repair brain damage. At the same time, the protective functions of astrocytes are also inhibited, which leads to BBB damage and allows a large number of cytotoxic substances to enter the brain and deteriorate damages [144].

## 6. NOS/NO Pathway and Inflammation (iNOS-Dominated)

iNOS is a primary mediator in inflammatory responses during cerebral ischemia and hypoxia [157]. The iNOS inhibitor aminoguanidine (AG) can significantly alleviate endothelial dysfunction and cognitive impairments in VaD rats. Researchers have also detected reductions in oxidative stress and inflammatory cytokines in these treated rats. AG has also been found to prevent demyelination and cerebral ischemia [158,159,160]. Moreover, compared with young and healthy individuals, a dramatically increased amount of iNOS mRNA expression is widely present in various somatic cells of the elderly, especially in VSMCs with vascular endothelial damage [111]. These results all indicated that the overexpression of iNOS is a crucial mechanism of VaD.

The expression of iNOS is regulated by the TLR4/NF-κB pathway (Figure 5). TLR4 is an innate immune receptor that is mainly distributed in the microglia and astrocytes of human brain. Its endogenous ligands can be released after cell injury, which activates the immune responses without the invasion of foreign pathogens. After that, a large amount of NF-κB is activated [161,162,163]. Thus, pathologically, hypoxia and persistent oxidative stress can trigger tissue damage and produce mass inflammatory cytokines such as TNF-α and TNF-β, which activate the TLR4/NF-κB pathway. Subsequently, it induces inflammation, which can continuously activate iNOS and produce excessive NO. Excessive iNOS-derived NO can react with ROS to generate ONOO^−^, which can cause oxidative damage to biomacromolecules (lipids, proteins, and DNA), leading to the deposition of cytotoxic substances. These cytotoxic substances can damage mitochondrial functions so that biological energy cannot meet the demand of neuron activities. It can also induce wide neural apoptosis and BBB destruction. Eventually, this damage leads to the inflammatory cascade and aggravates neurovascular dysfunctions [41,164].

Moreover, the massive secretion of inflammatory factors can stimulate the production of the Aβ amyloid protein, while the damaged CECs prevent Aβ-amyloid from being normally cleared, leading to deposition and triggering cerebral amyloid angiopathy [165]. The deposition of Aβ-amyloid can elevate the expression of p53 (a pro-apoptotic protein) but suppress Bcl-2 (an anti-apoptotic protein), which can be detected in VaD patients [166]. These excessive iNOS-derived NO can enhance the intracellular accumulation of p53, thereby initiating the necrosis process of neurons, CECs, and VSMCs [167,168,169,170,171]. Studies have found age-dependent neurovascular dysfunction in mice which with cerebral amyloid angiopathy [172].

Additionally, microglia and reactive astrocytes activated by inflammation can also be found in the chronic hypoperfusion area of a patient’s WM [173]. These cells can lead to the further release of inflammatory cytokines and adhesion molecules before activating the TLR4/NF-κB pathway—subsequently inducing an excessive activation of iNOS and producing NO in an amount that far exceeds physiological requirements. These excessive NO can induce demyelination, expose nerve cells to harmful substances, and finally induce neuroinflammation [64]. Additionally, the inflammatory factor TNF-α can bind to TNF receptor-1 in CECs to damage eNOS, thereby reducing the production of BDNF. It can lead to the inhibition of LTP formation and make neural stem cells differentiate into astrocytes, thus hindering neuronal regeneration and leading to the impairments of the NVU, which aggravates brain injuries [174,175,176,177].

Furthermore, it is worth mentioning that high-salt diets have been proven to cause a series of abnormalities in cerebral vascular and cognitive functions, such as high blood pressure [178]. Recently, a gut-brain axis was reported to be the underlying pathogenic mechanism. This axis can initiate the adaptive immune responses in the gut through a high-salt diet and activate endogenous iNOS. Then, the proliferation of T cells, especially Th17, is inhibited. While the level of circulating interleukin-17 (IL-17) is increased. IL-17 can activate the RhoA/RHO kinase pathway to enhance eNOS Thr495 phosphorylation, thereby inhibiting eNOS activity and leading to CEC dysfunction [78,179,180].

Ischemia, hypoxia, and continuous oxidative stress in the brain can induce inflammation, which can activate the TLR4/NF-κB pathway to produce a large number of inflammatory cytokines. These inflammatory cytokines can directly stimulate the massive expression of iNOS and lead to excessive NO production. They can also induce the deposition of Aβ amyloid protein, thereby increasing the expression of p53. These excessive NO can also enhance the intracellular accumulation of p53. Additionally, glial cells activated by inflammatory cytokines can also activate the TLR4/NF-κB pathway to active iNOS. These glial cells themselves can release inflammatory cytokines, forming a vicious cycle to increase inflammation. Moreover, inflammatory cytokines can reduce BDNF expression and then inhibit the pro-survival effect of neurons. Excessive NO can generate ONOO^−^, which can produce cytotoxicity to impair vasodilation and inhibit mitochondrial function. Additionally, these NO can inhibit the formation of LTP, thus reducing the regeneration of neurons that aggravate neurovascular dysfunction.

## 7. NOS/NO Pathway-Involved VaD Therapies

### 7.1. Physical Exercise

Many studies on VaD treatments have revealed a significant correlation between their efficacy and the NOS/NO pathway. In addition to the above-mentioned eNOS/NO-mediated BMMNC therapy [108], LRIC therapy [137], and LIPUS therapy [151], the neurovascular protection exerted by physical exercise is related to this pathway. Researchers have found that long-term exercise training can promote angiogenesis to improve cerebral ischemia in stroke mice models. They detected an increase in the eNOS activity and number of EPCs in these treated mice. However, eNOS-deficient mice treated with the same exercise regimen did not show these beneficial changes [106,181,182]. Moderate physical exercise has been shown to stimulate the expression of endogenous antioxidants such as SOD, GRx, and GR; reduce the activity of NADH oxidase; and reduce the production of inflammatory cytokines such as IL-6 and TNF-α, thereby limiting the spread of oxidative stress [183,184]. Moreover, physical exercise can also generate shear stress to stimulate CECs and then upregulate eNOS activity [185], thereby increasing CBF and triggering NVC to protect vascular and cognitive functions of the brain. Some studies have demonstrated that moderate aerobic exercise can improve cognitive abilities in patients with mild VCI [186,187].

### 7.2. Statins, Minocycline and NOS Inhibitors

Regarding drug therapy, statins—which are widely used in the treatment of stroke and atherosclerosis—have been a research hotspot. It was found that simvastatin can dramatically restore cerebrovascular function even in a late stage of VaD. This drug can increase eNOS activity by activating the PI3K/Akt pathway and increase the bioavailability of NO [188]. Another statin called lovastatin can inhibit the release of inflammatory cytokines, thereby inhibiting iNOS expression and avoiding the excessive production of NO [189]. More importantly, statins can also simultaneously upregulate eNOS activity and inhibit iNOS expression in an ischemic brain, hence enhancing vasodilation, reducing oxidative stress, preventing cell apoptosis and exerting neurovascular protection [152]. Another drug named minocycline has also been found to be able to upregulate eNOS activity and downregulate iNOS expression. This drug can pass the BBB, thereby inhibiting oxidative stress in the brain and alleviating cognitive impairments in VaD patients [190]. Meanwhile, selective NOS inhibitors such as AG and disubstituted indoline derivatives, have also been extensively studied recently. They have been shown to significantly improve symptoms in VaD mice [191]. Though these inhibitors still need to be tested in human clinical trials, they are undoubtedly very promising in the treatment of VaD.

### 7.3. Substrates and Cofactors Required for NO Synthesis

The substrates and cofactors required for NO synthesis have also been considered as potential drug targets for VaD therapies. BH_4_ is one of the most effective reducing agents in nature and is closely related to eNOS uncoupling. Bendall et al. [192] found that supplementation with BH_4_, the enhancement of BH_4_ biosynthesis, or the inhibition of BH_4_ oxidation in humans and animals with ischemia can reverse eNOS uncoupling and recover vascular endothelial injury.

Supplementation with L-Arg has also been found to contribute to CBF and nerve recovery, which can improve cognitive abilities in patients with cerebral ischemia [85,193]. In addition, L-Arg, as the precursor of NO, can serve as a potent biomarker for VaD. Its concentration in CSF and plasma is negatively correlated with infarct size, and its decrease can also reflect early neurological deterioration and poor prognosis [194]. However, Bahadoran et al. [195] found that the excessive intake of L-Arg from animals can cause adverse vascular effects, while intake from plants does not cause these and even exhibits vascular protection. This phenomenon may be related to the competitive transport of lysine and L-Arg in the cell. Lysine levels in animal proteins are higher than that of L-Arg, which may lead to the lower utilization of L-Arg in animals than in plants.

Additionally, the presence of inorganic nitrate, vitamin C, and polyphenols can also promote NO synthesis [92,196,197,198]. Currently, several small-scale studies have shown that changing patients’ dietary patterns (such as the Mediterranean diet) or feeding them foods rich in these substances (such as nitrate-rich spinach and beetroot) can increase CBF and improve endothelial dysfunction to varying degrees [196,197,198,199,200,201,202,203]. However, these studies were all carried out in strict clinical laboratory environments, and no more detailed tests such as 24-h dynamic monitoring were carried out to confirm how this change occurred. Therefore, the effect of VaD dietary therapy needs to be further studied through more large samples and clinical trials in daily life environment.

Additionally, asymmetric dimethylarginine (ADMA) is synthesized from the degradation of methylated proteins by L-Arg and is an effective endogenous non-selective NOS inhibitor. Oxidative stress can increase the level of ADMA by inactivating its degrading enzyme, dimethylarginine dimethylaminohydrolase (DDAH) [116,204,205], therefore suppressing eNOS activity and inducing CEC dysfunction. A large number of studies have confirmed the involvement of ADMA in the occurrence of a variety of vascular diseases [82,206,207,208]. Homoarginine, an L-Arg analog, can be used as an NOS substrate. It can increase the synthesis of NO and protect cerebrovascular functions. Recent studies have shown that circulating homoarginine concentration is negatively correlated with aging, while ADMA concentration is positively correlated with aging [208,209]. It has been proposed that changing the concentration of the two aforementioned substances could antagonize the adverse vascular effects caused by aging. Thus, it can be a new direction for the development of VaD therapies.

## 8. Conclusions

VaD has become a major public health threat all over the world. The NOS/NO pathway is involved in almost all the pathological mechanisms of VaD. NO has both protective and toxic tendencies. Under pathological conditions, aging and vascular risk factors lead to the inactivation of eNOS, which prevents the normal production of NO and reduces NO bioavailability. It triggers serious CEC dysfunction. At the same time, nNOS and iNOS are overactivated to produce an amount of NO that far exceeds the physiological requirements. This excessive NO leads to neurotoxicity and neuroinflammation. The pathological damage caused by these three NOS/NO pathways can affect each other and collectively lead to VaD.

Currently, many VaD treatments targeting the NOS/NO pathway are being studied. Meanwhile, supplementation with substrates and cofactors, such as BH_4_, L-Arg, inorganic nitrate, and vitamin C, which are required for NO synthesis, can also alleviate VaD symptoms. Thus, identifying the detailed pathogenic mechanism of VaD at the molecular level may provide more new ideas and breakthroughs for the development of effective treatment methods for VaD.

## Figures and Tables

**Figure 1 ijms-22-04540-f001:**
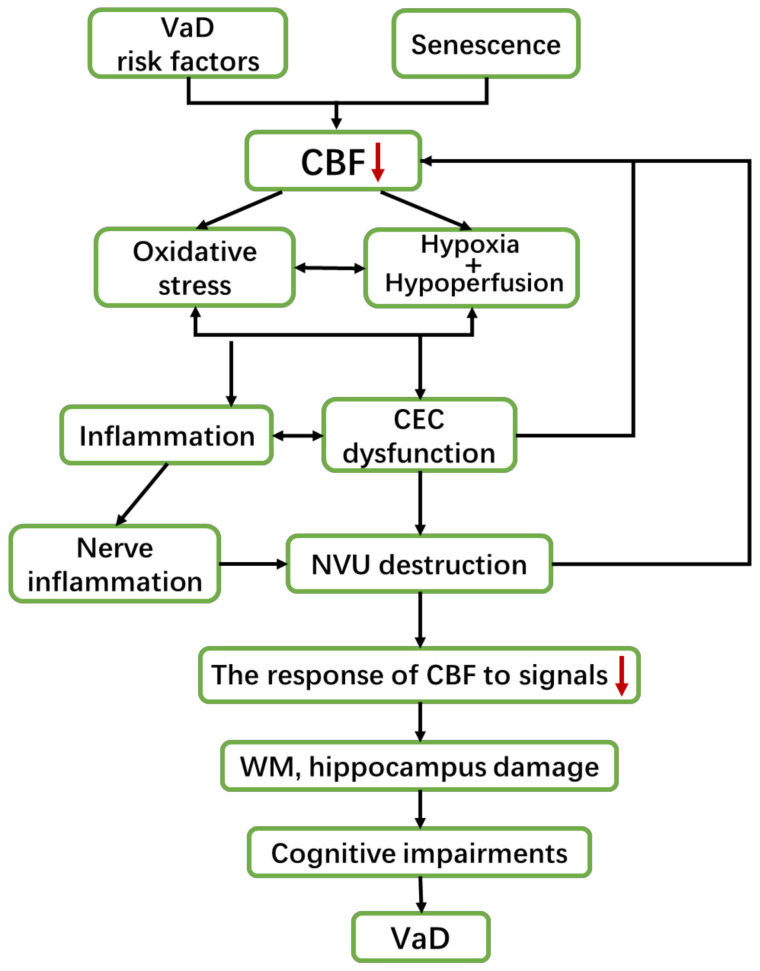
Overview of the pathological mechanism of VaD.

**Figure 2 ijms-22-04540-f002:**
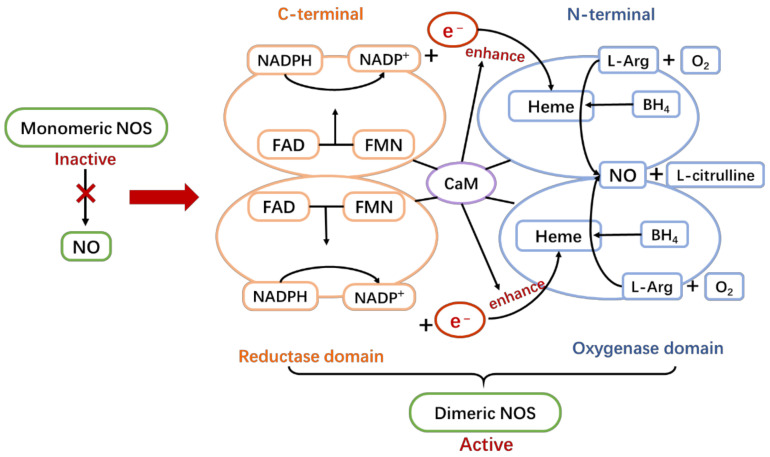
The active structure of NOS.

**Figure 3 ijms-22-04540-f003:**
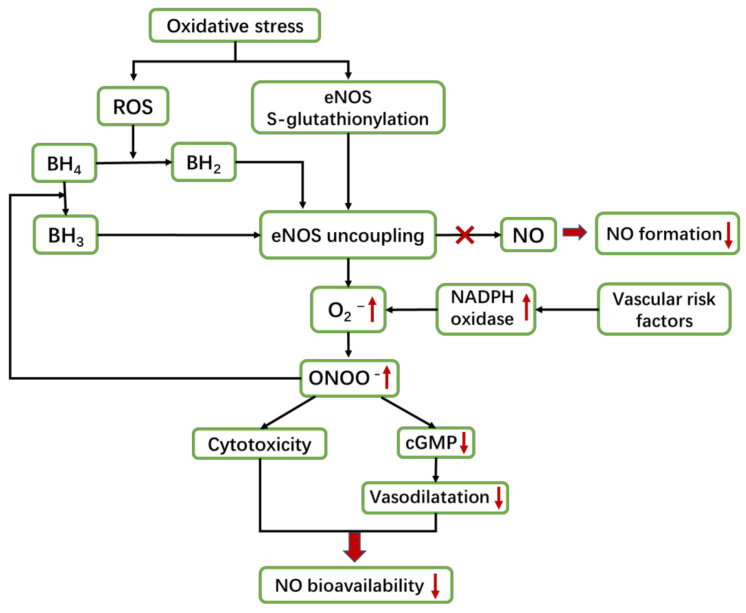
Mechanism of eNOS uncoupling.

**Figure 4 ijms-22-04540-f004:**
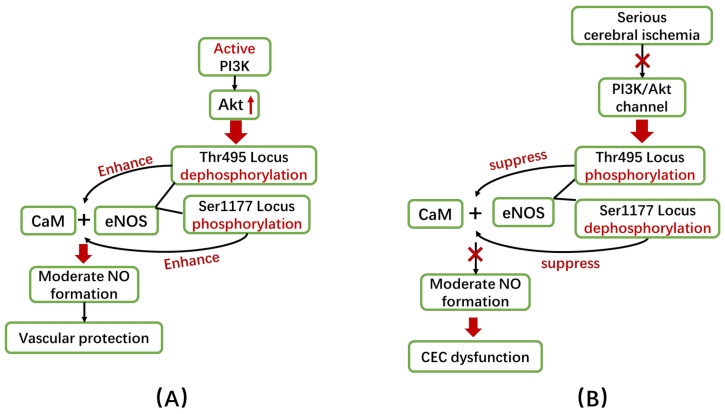
Alterations in eNOS phosphorylation state are involved in CEC dysfunction. (**A**). Under normal conditions, the PI3K/Akt pathway can promote the phosphorylation of eNOS at the Ser1177 locus and the dephosphorylation of the Thr495 locus, thereby enhancing the binding of eNOS to CaM. Thus, eNOS can be rapidly activated to produce adequate amounts of NO to protect vascular function. (**B**). However, during severe cerebral ischemia, the PI3K/Akt pathway is inhibited, leading to the altered phosphorylation of eNOS—i.e., dephosphorylation at the Ser1177 locus but phosphorylation at the Thr495 locus, which prevents eNOS from being activated and producing NO, thereby exacerbating the dysfunction of CECs.

**Figure 5 ijms-22-04540-f005:**
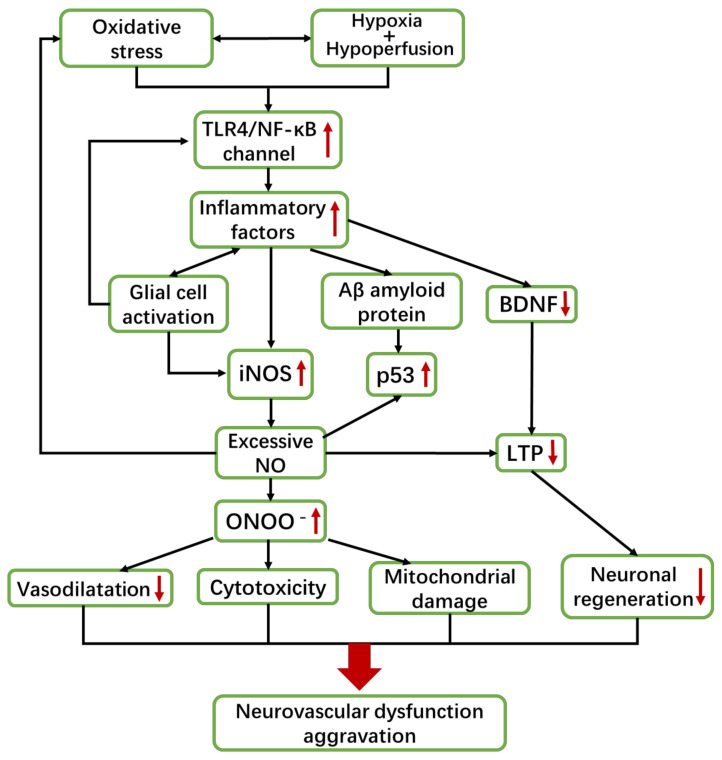
Inflammation leads to VaD pathogenesis by increasing the expression of iNOS.

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
