# Peer review of "The Roles of Nitric Oxide Synthase/Nitric Oxide Pathway in the Pathology of Vascular Dementia and Related Therapeutic Approaches"

_ijms, 2021, doi:10.3390/ijms22094540_

Round 1

Reviewer 1 Report

This manuscript review summarizes pathogenesis of vascular dementia (VaD) and how nitric oxide synthase / nitric oxide  mediates this process, and also proposes some potential therapeutic methods that target this pathway. This is an interest topic. However, weaknesses of the manuscript need to be improved by the author for possible publication in the International Journal of Molecular Sciences.

Specific comments:

  1. This review also summarizes some potential therapeutic methods that target this pathway, in the hope of inspiring novel ideas for VaD therapeutic approaches. However, the conclusion only pointed out that supplementation of substances and cofactors involved in NO synthesis has been found to alleviate VaD damages. Several important compounds studied for VaD have been omitted in the area.  
  2. There are many incorrect used punctuation, space, words, and sentences in the manuscript. For examples:

In p. 3, lines 111, …… mechanisms. which may….  should be corrected as …… mechanisms, which may….

In p. 8, lines 331, …… the the activities of … should be …… the activities of ….

In p. 9, lines 350, 351, 352, …… O2-can … should be ……… O2- can ….

                                       …… ONOO2-can … should be ……… ONOO2- can ….

Reviewer 2 Report

This is a comprehensive review, which adds to an extensive literature in the field associated with a high number of similar reviews. References and selection of data seem appropriate.

The concern is mostly on the design of the manuscript. A reader, who would approach the issue, might have hard time to gain real knowledge and would not be able to discriminate between established findings and suggestive data. What seems to be missing is a critical appraisal of the matter. Part of the problem comes from the very little attention paid to the methods behind the findings and quality of the source data. The manuscript is essentially a collection of abstracts. As a consequence, there are many repetitions.

An additional concern regards the style. On many occasions the sentences are not clear or the selection of words not appropriate to the underling meaning.

Just as examples: “Under such stimulus, the body chronic inflammation state for a long period, 68 which deteriorates” (line 68); “Oxidative stress is an unbalanced environment between oxidants 83 and antioxidants in the body”, line 83; “The neurological damages of VaD are developed from the deterioration of cerebro- 117 vascular diseases”, line 117; and many others

Round 2

Reviewer 2 Report

The manuscript is definitely improved and, as a whole, it provides the reader with a useful summary of the knowledge in the field. The manuscript still needs an extensive editing as far as it concerns the style and grammar. There are many sentences with wrong syntax and/or grammar.